# Causally Debiased Time-aware Recommendation

## ABSTRACT

Time-aware recommendation has been widely studied for modeling the user dynamic preference and a lot of models have been proposed. However, these models often overlook the fact that users may not behave evenly on the timeline, and observed datasets can be biased by user intrinsic preferences or previous recommender systems, leading to degraded model performance. We propose a causally debiased time-aware recommender framework to accurately learn user preference. We formulate the task of time-aware recommendation by a causal graph, identifying two types of biases on the item and time levels. To optimize the ideal unbiased learning objective, we propose a debiased framework based on the inverse propensity score (IPS) and extend it to the doubly robust method. Considering that the user preference can be diverse and complex, which may result in unmeasured confounders, we develop a sensitivity analysis method to obtain more accurate IPS. We theoretically draw a connection between the proposed method and the ideal learning objective, which to the best of our knowledge, is the first time in the research community. We conduct extensive experiments on three real-world datasets to demonstrate the effectiveness of our model. To promote this research direction, we have released our project at https://www-cdtr.github.io/.

## KEYWORDS

Time-aware recommendation, Collaborative filtering, Counterfactual, Causal inference.

**ACM Reference Format:**
Anonymous Author(s). 2024. Causally Debiased Time-aware Recommendation. In *Proceedings of Proceedings of the ACM Web Conference 2024 (WWW '24)*. ACM, New York, NY, USA, 11 pages. https://doi.org/XXXXXXX.XXXXX

## 1 INTRODUCTION

The key of a recommender system lies in the accurate understanding of the user preference. In real-world scenarios, the user preference usually exhibits dynamic properties. For example, in the e-commerce recommendation, the users may purchase more items when they are on sale. In the movie recommendation, one may give higher ratings at weekends due to the holiday mood. To capture the dynamic nature of the user preference, researchers have designed a lot of promising time-aware recommender models. For example, TimeSVD++ [23] incorporates the time information into the matrix factorization methods. BPTF [42] leverages tensor factorization to captures the user, item and time interactions.

*WWW '24, May 13–17, 2024, SINGAPORE*
© 2024 Association for Computing Machinery.
ACM ISBN 978-1-4503-XXXX-X/18/06...$15.00
https://doi.org/XXXXXXX.XXXXX

While the above models have achieved remarkable successes, an important problem has been largely ignored, that is, the observed datasets can be biased by the user intrinsic preference or the previous recommender models. As exampled in Figure 1, the user is a fan of science-fiction. Thus in her watching records, we may observe much more sci-fi movies than other types of films. If the model is learned based on such dataset, the performance can be unsatisfying if different types of movies are evaluated with the same weights (*i.e.*, in an unbiased manner) [4]. From the interaction time perspective, due to the watching habit, the user watches more movies at weekends, while on the working days, the interactions are quite sparse. With these records, the model will learn more about the user weekend behavior patterns (*e.g.*, tending to give higher ratings), which may not work well for the working days. However, an ideal temporal model should accurately estimate the user preference for all the interaction times. The above item- and time-level biases jointly influence the observed datasets, and the models learned based on them may be biased and not perform well on the disadvantaged items/times.

To alleviate the above problem, in this paper, we propose a **c**ausally **d**ebiased **t**ime-aware **r**ecommender framework (called **CDTR** for short). In particular, we formulate the task of time-aware recommendation by a causal graph, based on which we analyze the causes of the item- and time-level biases. To correct these biases, we adjust the training samples based on the inverse propensity score (IPS) [31]. Although this seems to be a straightforward idea, there are many challenges. To begin with, debiased recommendation has been extensively studied before. However, previous work only focused on the item-level bias. In our problem, there are two types of biases, that is, the item- and time-level biases. How to jointly model and correct them needs careful designs. Then, accurately estimating IPS for a recommender system is not easy, since the datasets can be quite sparse and noisy. Previous work usually alleviated this problem by building doubly robust (DR) models [38]. However, how to extend these models to the time-aware settings is not clear. At last, due to the complex user preferences, there can be many unmeasured confounders which are not recorded in the dataset. How to handle them is also a challenge.

To overcome the above challenges, we firstly define the ideal unbiased learning objective. Then, we deploy two independent models to estimate the propensity scores for the items and times, respectively, which are expected to flexibly capture the item/time observational patterns. Based on these propensity scores, we design a time-aware debiased recommender framework, and also extend it to the doubly robust method. To reveal the rationalities of these methods, we theoretically prove that they are unbiased to the ideal objective. To capture the unmeasured confounders, we develop a sensitivity analysis method [9] by quantitatively indicating the degree of the unconfoundedness. Basically, we first find the "worst-case IPS" in a range to maximize the loss function, which is then minimized to learn the parameters by fixing the IPS. For the proposed method, we theoretically prove that the objective is an upper

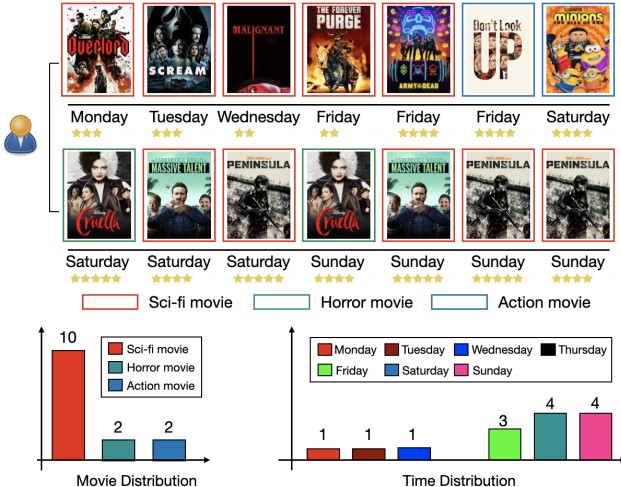

**Figure 1: Motivating example. The upper sub-figure presents the user watching records, including the viewed movies, interaction times and user-posted ratings. The bottom sub-figure shows the uneven distributions of the viewed movies and interaction times. (best view in color)**

bound of the ideal objective. To demonstrate the effectiveness of our framework, we apply it to different time-aware recommender models, and conduct extensive experiments based on three real-world datasets.

Notably, we have noticed that there is a pioneer work [16] on dynamic debiased recommender model. However, our paper has significant differences. To begin with, our model is derived from the causal perspective. In particular, we formulate the task of time-aware recommendation by a causal graph, which enables us to clearly understand the item- and time-level biases, and be aware of the potential unmeasured confounders. Then, we propose a doubly robust method to improve the estimation accuracy. At last, we provide a series of theoretical understandings of our model.

The main contributions of this paper can be summarized as follows:
- We propose to build a debiased time-aware recommender framework from the causal perspective.
- To achieve this goal, we design an IPS based model, and extend it by building a doubly robust method and explicitly modeling the unmeasured confounders.
- We provide a series of theoretical analysis on the designed models.
- Extensive experiments have been conducted to demonstrate the effectiveness of our models, and we have released our project at https://www-cdtr.github.io/.

## 2 RELATED WORK

### 2.1 Debiased Recommendation

In recent years, the topic of debiased recommendation has gained significant attention due to the realization that many recommendation systems suffer from biases that can degrade recommendation performance. Recommendation datasets usually contain different types of biases because the data are observational rather than experimental. To remove the data biases, recent years have witnessed a large amount of debiased recommender frameworks. Schnabel et al. [31] proposes a method that removes the item exposure bias by incorporating the inverse propensity score into the traditional matrix factorization model. To estimate propensity score more accurately, [44] develops an influence function based method to correct biases in the recommender system. Another method proposed by [38] combines error imputation and inverse propensity score to design a doubly robust method for unbiased recommendation. Subsequently, researchers have focused on optimizing the doubly robust method further, using various techniques to reduce the bias and variance. To this end, [7] formulates various doubly robust methods into a unified and general paradigm and further proposes to reduce bias and achieve a better trade-off between the bias and variance.

Unmeasured confounding is a common problem in the field of recommender systems, where there may be hidden factors that influence user preferences and item ratings, but which are not captured by the available data. In response, researchers have developed a variety of techniques to address the issue of unmeasured confounding. The first line is to model and estimate the effect of unmeasured confounding using statistical and machine learning methods. [39] uses neural networks to model the impact of unobserved latent confounding on recommendation results and incorporates it into the optimization process. [46] proposes iDCF, a general deconfounded recommender framework that applies proximal causal inference to infer the unmeasured confounders and identify the counterfactual feedback. Another line is using sensitivity analysis to learn a robust model that considers unmeasured confounders within a bound. In the field of causal inference, [19] introduces confounding-robust policy that takes into account possible unobserved confounding. Similarly, in [9], the authors make the assumption that the impact of observed confounding on the recommender system is limited. They use sensitivity analysis to estimate this influence range and then leverage adversarial learning to construct a recommendation system that is robust to unobserved confounding within this limit.

Most of the above models only consider the biases from the item perspective. However, in our framework, we jointly consider the item- and time-level biases. We have also improved the robustness of our recommendation system to unobserved confounding by conducting sensitivity analysis from both the perspectives of item and time. The most similar work to our paper is [16], but as mentioned before, there are significant differences on the modeling perspective, framework components and theoretical analysis.

### 2.2 Dynamic Recommendation

Dynamic recommendation can be generally divided into two classes according to how the time information is utilized [3, 6, 34]. In the following, we briefly introduce each of them.

**Time-dependent recommendation**. In recent times, this particular type of recommendation has gained significant attention from the research community. In this type of models, the time is leveraged to chronologically sort the items, and the models focus more on the item correlations along the timeline. Sequential recommendation [36], session based recommendation [35] and next-basket recommendation [43] all belong to this class. For example,

He et al. [13] propose Fossil assuming that the user behaviors can be approximated as a Markov process, where the current action is only influenced by the most recent decision. Fossil combines FISM [18] and Markov chain to model, where FISM is used to model long-term interest, and Markov chain is used to model short-term interest. In recent years, researchers have started to explore the use of RNNs in the field of recommendation systems, particularly for sequential recommendation tasks. Hidasi et al. [15] propose a session-based recommendation method based on GRUs, achieving state-of- the art performance with fewer parameters. [20] uses self-attention to model the influence of the items interacted long before. [33] leverages the bidirectional Transformers to model the user behaviors. In addition, many models incorporate the absolute time information to highlight the importance of the time interval between successive items for the user decisions [45].

**Time-aware recommendation**. In this type of methods, the time is used as a contextual information, which will directly influence the model predictions. For example, [23] combines the time information with the matrix factorization model, and the user and item properties are represented by temporal latent vectors. [10] imposes exponential decay rates on the older user behaviors to make more reasonable estimations. [42] is a tensor factorization method, where the time information is regarded as an additional dimension to capture the user dynamic preference. [6] leverages deep convolutional neural network to capture the nonlinear correlations between the time and user-item.

Our paper targets at time-aware recommendation. However, different from the previous work, we do not aim to develop an additional model to better incorporate the time information. We reveal a fundamental problem, that is, the observed time information can be biased due to the user selection preferences. In addition, we build a debiased framework, which can be applied to most of the previous time-aware recommender models.

## 3 PROBLEM FORMULATION

Our paper focuses on time-aware recommendation. Time-aware recommendation is motivated by the dynamic nature of the user preference [3]. We have used multiple symbols and abbreviations in this paper, and their meanings are presented in Table 1.

Formally, suppose we have a user set $\mathcal{U}$ and an item set $\mathcal{V}$. The interactions between the users and items are defined as

$$O = \{(u, v, t, r_{uvt}) | u \in \mathcal{U}, v \in \mathcal{V}, t \in \mathcal{T}\},$$

where each element $(u, v, t, r_{uvt})$ means that user $u$ has interacted with item $v$ at time $t$ with feedback $r_{uvt}$. $\mathcal{T}$ is the set of all possible interaction times. $r_{uvt}$ can be either explicit feedback like the rating from the user to the item, or implicit feedback, such as whether the user has clicked or purchased the item[1]. Based on $\{\mathcal{U}, \mathcal{V}, \mathcal{T}, O\}$, time-aware recommendation aims to learn a model $f$, such that for a given user-item pair $(u, v)$ and an interaction time $t$, $f$ can accurately estimate the likeness from the user to the item at time $t$.

In order to learn $f$, the following time-aware objective is usually optimized:

$$L = \frac{1}{|O|} \sum_{(u,v,t,r_{uvt}) \in O} \delta(f(u, v, t), r_{uvt}), \tag{1}$$

---

[1]In practice, the implicit feedback is usually converted to a 0-1 value.

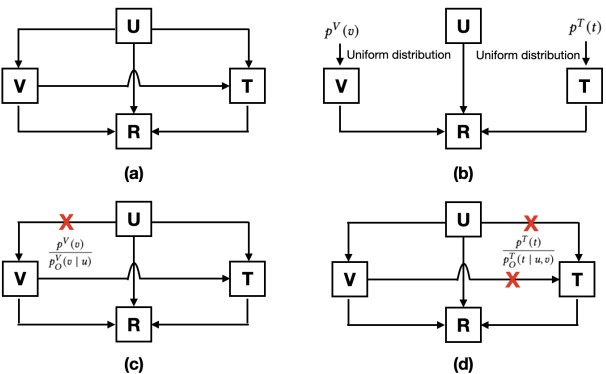

**Figure 2: (a) The causal graph for generating the observed dataset. (b) The causal graph for generating the dataset for training the ideal learning objective. The effects of the inverse propensity scores on the causal graph of (a) are shown in (c) and (d).**

where $\delta$ is the loss for a specific sample. For the explicit feedback, $\delta$ can be implemented with a regression loss, such as the mean squared error [12]. For the implicit feedback, the task is usually converted to a classification problem, and $\delta$ can be specified with the binary cross entropy loss [37]. In the past few decades, people have devoted a lot of effort to designing $f$ for better fitting the observed datasets [14, 20, 33]. However, the observed datasets can be skewed by the user self-selection preferences or the recommender systems used, which may bias the learned models.

**Causal understanding of time-aware recommendation**. In order to more deeply understand the data biases in time-aware recommendation, we provide a causality analysis of this task. To begin with, we present a causal graph to describe the observed data generation process in Figure 2(a), where $U$, $V$, $T$ and $R$ represent the user, item, interaction time and feedback, respectively. The rationalities of this causal graph are explained as follows:

• $U, V, T \rightarrow R$: these edges encode the assumption that the feedback is jointly determined by the user, item and interaction time, which is the basis for all the time-aware recommender models.

• $U \rightarrow V$: this edge represents that the observation of the item is influenced by the user. For example, if the item is actively selected by the user, then the user intrinsic preference determines which item is observed (*i.e.*, self-selection bias). If the item is recommended by the previous systems, then, since most recommender models are personalized, the observed item is influenced by the user (*i.e.*, selection bias). Both of the self-selection and exposure biases can be abstracted by the edge $U \rightarrow V$.

• $U \rightarrow T, V \rightarrow T$: these edges mean that the interaction time is jointly influenced by the user and item. On one hand, it is intuitive that different users may click/purchase the same item at various times. On the other hand, for the same user, different items also determine their own interaction times. For example, one may purchase T-shit in summer, but for the cotton coat, the interaction time is more likely to be the winter.

**The ideal learning objective and causal graph**. Intuitively, an ideal time-aware recommender model should accurately estimate the user feedback on all items at all times, which corresponds to

the following learning objective:

$$
\begin{aligned}
L_{\text{ideal}} &= \frac{1}{|\mathcal{U}|} \sum_{u \in \mathcal{U}} \frac{1}{|\mathcal{V}|} \sum_{v \in \mathcal{V}} \frac{1}{|\mathcal{T}|} \sum_{t \in \mathcal{T}} \delta(f(u,v,t), r_{uvt}) \\
&= \frac{1}{|\mathcal{U}|} \sum_{u \in \mathcal{U}} E_{v \sim p^V(v)} [E_{t \sim p^T(t)} [\delta(f(u,v,t), r_{uvt})]],
\end{aligned}
\tag{2}
$$

where the samples iterate all the users, items and times. Here, we follow the previous work [1, 25, 28] to discretize the time, and all possible times are collected in $\mathcal{T}$. $p^V(v) = \frac{1}{|\mathcal{V}|}$ ($\forall v \in \mathcal{V}$) and $p^T(t) = \frac{1}{|\mathcal{T}|}$ ($\forall t \in \mathcal{T}$) are the uniform distributions on the item and time sets, respectively.

To optimize objective (2), the datasets should be generated according to the ideal causal graph presented in Figure 2(b), where the item and time are independent with the user, and uniformly observed. In practice, obtaining such datasets needs random online experiments, which can be too expensive or even infeasible [8].

By comparing the causal graph for generating the observed dataset (*i.e.*, Figure 2(a)) with the ideal one (*i.e.*, Figure 2(b)), we can find two major differences: on one hand, the item observation in Figure 2(a) is influenced by the user, thus there will be more user favored items in the observed dataset than those in the ideal dataset (generated based on Figure 2(b)). This is the basic reason for the item-level bias. On the other hand, the interaction time in Figure 2(a) is influenced by the user and item, which makes the observational time distribution not uniform as required by the ideal dataset, leading to the time-level bias. The existence of the item- and time-level biases makes the traditional objective (1) deviated from the ideal one. In this paper, we would like to design a debiased framework which is more aligned with the ideal objective.

## 4 THE CDTR MODEL

To achieve the above goal, in this section, we propose a debiased time-aware recommender framework. In particular, we firstly design an IPS method to correct the item- and time-level biases simultaneously. Then we extend the IPS method to the doubly robust model. At last, we capture the unmeasured confounders by a sensitivity analysis method, which facilitates more accurate IPS estimation.

### 4.1 The Temporal IPS Method

IPS is a widely-used and effective method for correcting the data bias in recommendation. [4] This method adjusts the training samples by assigning importance weights based on the inverse of their probability of observation. Instances with smaller observation probabilities are assigned higher weights, whereas more frequently observed samples are assigned lower weights. Although previous research has shown that IPS can effectively address item-level biases, these methods are not suitable for handling the biases at both the item and time levels. To simultaneously handle the item- and time-level biases, we design the following objective:

$$
L_{\text{IPS}} = \frac{1}{|\mathcal{U}|} \sum_{u \in \mathcal{U}} \left\{ \frac{1}{|\mathcal{V}_u|} \sum_{v \in \mathcal{V}_u} \frac{p^V(v)}{p_O^V(v|u)} \left[ \frac{1}{|\mathcal{T}_{uv}|} \sum_{t \in \mathcal{T}_{uv}} \frac{p^T(t)}{p_O^T(t|u,v)} \delta_{uvt} \right] \right\},
$$

where $\delta_{uvt}$ is short for $\delta(f(u,v,t), r_{uvt})$. $\mathcal{V}_u$ is the set of items interacted by user $u$. $\mathcal{T}_{uv}$ is the set of interaction times[2] for the user-item pair $(u,v)$. $p_O^V(v|u)$ represents the probability of user $u$ selecting item $v$, while $p_O^T(t|u,v)$ represents the probability of time $t$ being observed for the user-item pair $(u,v)$.

Based on the above design, we have the following theory.

**Theorem 1** (Unbiasedness of $L_{\text{IPS}}$). *$L_{\text{IPS}}$ is an unbiased estimator for $L_{\text{ideal}}$, that is, $E[L_{\text{IPS}}] = L_{\text{ideal}}$.*

The proof of this theorem is presented in the Appendix. The above theorem suggests that we can unbiasedly estimate $L_{\text{ideal}}$ based on the observed dataset. From the causal perspective, the weight $\frac{p^V(v)}{p_O^V}$ adjusts the distribution of $V$ to be uniform, which actually cuts down the relation between $U$ and $V$ (see Figure 2(c)). Similarly, by introducing the weight $\frac{p^T(t)}{p_O^T}$, we aim to make the variable $T$ uniformly distributed on the time set $\mathcal{T}$, which blocks the edges $U \to T$ and $V \to T$ in Figure 2(d).

### 4.2 The Temporal Doubly Robust Method

In the above section, we have designed an unbiased estimator for the ideal objective. However, the real propensity scores $p_O^V(v|u)$ and $p_O^T(t|u,v)$ are not accessible, which need to be estimated. In practice, estimating the propensity score may suffer from the low accuracy and high variance [11]. Previous work usually alleviate these problems using doubly robust (DR) methods [38]. In this section, we extend this idea to the time-aware recommendation, and propose a temporal doubly robust method. In particular, we firstly introduce an imputation model $\hat{\delta}_{uvt}$ to estimate the loss for all the user-item-time triplets, and then correct it by an unbiased deviation term $\triangle_{uvt} = \delta_{uvt} - \hat{\delta}_{uvt}$ based on the estimated propensity scores. The final objective for our doubly robust method is:

$$
\begin{aligned}
L_{\text{DR}} = &\frac{1}{|\mathcal{U}|} \sum_{u \in \mathcal{U}} \left\{ \frac{1}{|\mathcal{V}_u|} \sum_{v \in \mathcal{V}_u} \frac{p^V(v)}{\hat{p}_O^V(v|u)} \left[ \frac{1}{|\mathcal{T}_{uv}|} \sum_{t \in \mathcal{T}_{uv}} \frac{p^T(t)}{\hat{p}_O^T(t|u,v)} \triangle_{uvt} \right] \right\} \\
&+ \frac{1}{|\mathcal{U}||\mathcal{V}||\mathcal{T}|} \sum_{u \in \mathcal{U}} \sum_{v' \in \mathcal{V}} \sum_{t' \in \mathcal{T}} \hat{\delta}_{uv't'},
\end{aligned}
$$

where $\hat{p}_O^V(v|u)$ and $\hat{p}_O^T(t|u,v)$ are the estimated propensity scores. The imputation model $\hat{\delta}_{uvt}$ is optimized based on the following objective:

$$
L_{\text{imp}} = \frac{1}{|\mathcal{U}|} \sum_{u \in \mathcal{U}} \sum_{v \in \mathcal{V}_u} p_{uv} \sum_{t \in \mathcal{T}_{uv}} p_{uvt} (\delta_{uvt} - \hat{\delta}_{uvt})^2,
$$

where $p_{uv} = \frac{p^V(v)}{|\mathcal{V}_u| \hat{p}_O^V(v|u)}$ and $p_{uvt} = \frac{p^T(t)}{|\mathcal{T}_{uv}| \hat{p}_O^T(t|u,v)}$.

By the inverse propensity scores, we aim to make the learned imputation model unbiased. The parameters of $\delta_{uvt}$ and $\hat{\delta}_{uvt}$ are alternatively optimized based on $L_{\text{DR}}$ and $L_{\text{imp}}$, respectively.

In the following, we discuss the unbiasedness and variance of $L_{\text{DR}}$. The detailed derivation can be found in Appendix.

---

[2]In most recommendation datasets, each user interacts with an item at most once. Thus, $|\mathcal{T}_{uv}| = 1$ in most cases.

**Theorem 2** (The Bias of $L_{DR}$). *Suppose $p_O^V(v|u)$ and $\hat{p}_O^V(v|u)$ are the real and estimated propensity scores for the item-level bias; $p_O^T(t|u,v)$ and $\hat{p}_O^T(t|u,v)$ are the real and estimated propensity scores for the time-level bias; $\triangle_{uvt} = \delta_{uvt} - \hat{\delta}_{uvt}$ is the error between the real and estimated losses. We define the bias of $L_{DR}$ from $L_{ideal}$ as:*

$$Bias(L_{DR}) = |\mathrm{E}\,[L_{DR}] - L_{ideal}|,$$

*then, we have:*

$$Bias(L_{DR}) = \left| \frac{1}{|\mathcal{U}||\mathcal{V}||\mathcal{T}|} \sum_{u \in \mathcal{U}} \sum_{v \in \mathcal{V}} \sum_{t \in \mathcal{T}} \triangle_{uvt} \left( \frac{s_{uvt} - \hat{s}_{uvt}}{\hat{s}_{uvt}} \right) \right|,$$

*where*

$$s_{uvt} = p_O^V(v|u)p_O^T(t|u,v), \hat{s}_{uvt} = \hat{p}_O^V(v|u)\hat{p}_O^T(t|u,v).$$

This theorem suggests that $L_{DR}$ is an unbiased estimator of $L_{ideal}$ when the propensity scores are accurately estimated (*i.e.*, $s_{uvt} = \hat{s}_{uvt}$) or the imputation model can perfectly predict the real loss (*i.e.*, $\delta_{uvt} = \hat{\delta}_{uvt}$). Comparing with $L_{IPS}$, $L_{DR}$ provides additional opportunities to achieve the unbiased estimator, that is, even if the propensity scores are wrongly predicted, we may also rely on an accurate enough imputation model.

**Theorem 3** (The Variance of $L_{DR}$). *We define*

$$w_{uvt} = \frac{p^V(v)p^T(t)}{|\mathcal{V}_u||\mathcal{T}_{uv}|\hat{p}_O^V(v|u)\hat{p}_O^T(t|u,v)},$$

*and let $X_{uvt} = w_{uvt}\delta_{uvt}$ and $Y_{uvt} = w_{uvt}\hat{\delta}_{uvt}$, then we have:*

$$Var(L_{DR}) = \frac{1}{|\mathcal{U}|^2} \sum_{u \in \mathcal{U}} \sum_{v \in \mathcal{V}_u} \sum_{t \in \mathcal{T}_{uv}} Var(X_{uvt} - Y_{uvt}),$$

*and $Var(L_{DR}) < Var(L_{IPS})$.*

## 4.3 Unmeasured Confounder Modeling

In the above sections, we design and improve the IPS based framework by assuming that the causal graph in Figure 2(a) is causally sufficient, that is, there are no unmeasured confounders. However, in reality, the factors that influence the selections of the items and times are quite complex and diverse, and there can be many factors which are not recorded in the datasets, for example, the user emotions and item promotions. These factors invalidate the causally sufficient assumption. [9]

To alleviate this problem, we propose to relax this assumption by explicitly modeling the unmeasured confounders. A straightforward method to capture the unmeasured confounders is designing a parametric model to infer them [2, 26]. However, without sufficient prior knowledge, the model can be incorrectly specified, which may lower the final performance. To overcome this weakness, we introduce a non-parametric sensitivity analysis method [32], where we only quantify the strength of the unmeasured confounders without imposing any model assumptions. More specifically, for the item-level bias, we estimate the propensity score is $\hat{p}_{uv} = \hat{p}_O^V(v|u) = \frac{\exp(g_{uv})}{\sum_{v' \in \mathcal{V}} \exp(g_{uv'})}$, where $g_{uv}$ can be any scoring function for the user-item pair $(u, v)$.

Suppose the unmeasured confounders which influence the user-item selections are represented by $h_{uv}$, then in $\hat{p}_O^V(v|u)$, the prediction of the item should also be dependent on $h_{uv}$. Similar to the

previous work [9], we consider an additive model for deriving the propensity score based on the unmeasured confounders, that is $\bar{p}_{uv} = \bar{p}_O^V(v|u, h_{uv}) = \frac{\exp(g_{uv}+\beta(h_{uv}))}{\sum_{v' \in \mathcal{V}} \exp(g_{uv'}+\beta(h_{uv'}))}$, where $\beta$ can be any function projecting $h_{uv}$ into a scalar, indicating the strength of the unmeasured confounders. Instead of specifying $\beta$ with a parametric model, we only indicate the value space of $\beta(h_{uv})$, and optimize $L_{IPS}$ and $L_{DR}$ based on the uncertainty set of $\bar{p}_{uv}$ represented by $\hat{p}_{uv}$.

In particular, suppose $\beta(h_{uv}) \in [a, b]$, then we have: $\bar{p}_{uv} \in [\frac{1}{[1+(\frac{1}{\hat{p}_{uv}}-1)\Gamma_{uv}]}, \frac{1}{[1+(\frac{1}{\hat{p}_{uv}}-1)\Gamma_{uv}^{-1}]}] \triangleq A_{uv}$, where $\Gamma_{uv} = e^{b-a}$ is a hyper-parameter larger than 1.

Similarly, for the time-level bias, we estimate the original IPS and the one considering the unmeasured confounders as follows: $\hat{p}_{uvt} = \hat{p}_O^T(t|u,v) = \frac{\exp(g_{uvt}^1)}{\sum_{t' \in \mathcal{T}} \exp(g_{uvt'}^1)}$ and $\bar{p}_{uvt} = \bar{p}_O^T(t|u,v,h_{uvt}) = \frac{\exp(g_{uvt}^1+\beta'(h_{uvt}^1))}{\sum_{v' \in \mathcal{V}} \exp(g_{uvt'}^1+\beta'(h_{uvt'}^1))}$, where $g^1$ and $\beta'$ can be any functions, and $h_{uvt}^1$ is the representation of the unmeasured confounders which influence the selection of the time given a user-item pair.

Suppose $\beta(h_{uvt}^1) \in [c, d]$, then, we have: $\bar{p}_{uvt} \in [\frac{1}{[1+(\frac{1}{\hat{p}_{uvt}}-1)\Gamma_{uvt}]}, \frac{1}{[1+(\frac{1}{\hat{p}_{uvt}}-1)\Gamma_{uvt}^{-1}]}] \triangleq B_{uvt}$, where $\Gamma_{uvt} = e^{d-c}$ is a hyper-parameter.

The key of our sensitivity analysis method is optimizing $L_{IPS}$ and $L_{DR}$ based on the uncertainty sets of $\bar{p}_{uv}$ and $\bar{p}_{uvt}$. In particular, we use adversarial learning to optimize the maximum loss induced by the uncertainty sets, which helps to remove the unstable factors result from the unmeasured confounders, and lead to more robust optimization. Formally, we improve $L_{IPS}$ and $L_{DR}$ as follows:

$$L_{\text{IPS-UM}} = \max_{\substack{\bar{p}_{uv} \in A_{uv}, \\ \bar{p}_{uvt} \in B_{uvt}}} C \sum_{u \in \mathcal{U}} \left\{ \frac{1}{|\mathcal{V}_u|} \sum_{v \in \mathcal{V}_u} \frac{1}{\bar{p}_{uv}} \left[ \frac{1}{|\mathcal{T}_{uv}|} \sum_{t \in \mathcal{T}_{uv}} \frac{1}{\bar{p}_{uvt}} \delta_{uvt} \right] \right\},$$

$$L_{\text{DR-UM}} = \max_{\substack{\bar{p}_{uv} \in A_{uv}, \\ \bar{p}_{uvt} \in B_{uvt}}} C \sum_{u \in \mathcal{U}} \left\{ \frac{1}{|\mathcal{V}_u|} \sum_{v \in \mathcal{V}_u} \frac{1}{\bar{p}_{uv}} \left[ \frac{1}{|\mathcal{T}_{uv}|} \sum_{t \in \mathcal{T}_{uv}} \frac{1}{\bar{p}_{uvt}} \triangle_{uvt} \right] \right\}$$
$$+ C \sum_{u \in \mathcal{U}} \sum_{v' \in \mathcal{V}} \sum_{t' \in \mathcal{T}} \hat{\delta}_{uv't'},$$

where $C = \frac{1}{|\mathcal{U}||\mathcal{V}||\mathcal{T}|}$. To more accurately debias the imputation model $\hat{\delta}_{uvt}$, we also learn it based on the uncertainty sets of $\bar{p}_{uv}$ and $\bar{p}_{uvt}$, which revises $L_{imp}$ as follows:

$$L_{\text{imp-UM}} = \max_{\substack{\bar{p}_{uv} \in A_{uv}, \\ \bar{p}_{uvt} \in B_{uvt}}} \frac{1}{|\mathcal{U}|} \sum_{u \in \mathcal{U}} \sum_{v \in \mathcal{V}_u} p_{uv} \sum_{t \in \mathcal{T}_{uv}} p_{uvt} (\delta_{uvt} - \hat{\delta}_{uvt})^2,$$

where $p_{uv} = \frac{p^V(v)}{|\mathcal{V}_u|\bar{p}_{uv}}$ and $p_{uvt} = \frac{p^T(t)}{|\mathcal{T}_{uv}|\bar{p}_{uvt}}$.

Basically, $L_{\text{IPS-UM}}$ and $L_{\text{DR-UM}}$ learn the model based on the IPS uncertainty sets. We find that such type of optimization has close connections with the generalization error bound of the ideal objective $L_{ideal}$. We focus our analysis on $L_{\text{IPS-UM}}$, and the results can be easily extended to $L_{\text{DR-UM}}$.

**Theorem 4.** *Suppose $p_{uv}^*$ and $p_{uvt}^*$ are the ground truths of the propensity scores, which are unknown, $|\delta_{uvt}| \le \kappa$. All the propensity*

scores are bounded in the range of $[s_1, s_2]$, where $0 < s_1 < s_2 < 1$. If we set $\lambda_1 = \lambda_2 = \frac{\kappa s_2^2(1-s_1)}{1-s_1+s_2}$, then,

$$L_{ideal} \leq L_{\text{IPS-UM}} + Const, \qquad (3)$$

where Const is a constant.

This theorem suggests that by optimizing $L_{\text{IPS-UM}}$, we actually lower the upper bound of the ideal objective, which reveals the rationality of our model in theory. We present the proof of this theorem in the Appendix.

## 4.4 Model Specification

In this section, we detail the specifications of functions. To begin with, $\delta_{uvt}$ is implemented with the mean squared error loss. The recommender model in $\hat{\delta}_{uvt}$ is the same as the one in $\delta_{uvt}$, but with independent parameters. To predict the propensity scores, we firstly introduce two embedding matrices $E^U \in R^{\mathcal{U} \times d}$ and $E^V \in R^{\mathcal{V} \times d}$ for the users and items, respectively. Suppose $e_u^U \in R^d$ and $e_v^V \in R^d$ are the $u$-th and $v$-th columns of $E^U$ and $E^V$, representing the embeddings of the user $u$ and item $v$. Then, we have $\hat{p}_{uv} = [\text{SOFTMAX}(E^V(e_u^U)^T)]_v$, where $[\cdot]_v$ indicates the $v$-th element of a vector and SOFTMAX is the softmax operator. $\hat{p}_{uvt} = [\text{SOFTMAX}(\text{MLP}([e_u^U; e_v^V]))]_t$, where $[\cdot; \cdot]$ is the concatenation operation and MLP projects the input into a $|\mathcal{T}|$-dimension vector. We estimate $\overline{p}_{uv}$ (or $\overline{p}_{uvt}$) based on a MLP, where the input is the concatenation of the user-item (or user-item-time) embeddings, and the output is a scalar. Since different $\overline{p}_{uv}$'s (or $\overline{p}_{uvt}$'s) are predicted by a unified framework, they are actually dependent.

## 5 EXPERIMENTS

### 5.1 Experiment Setup

**Datasets.** Our experiments are conducted on three real-world datasets:

- **Movielens-1M(ML-1M)**[3] is a commonly used recommendation dataset, which contains the user-movie interactions and the time information.
- **Amazon**[4] is an e-commerce dataset, which includes the user purchasing behaviors on the products.
- **Food**[5] is a dataset containing the user preferences on the foods, which is collected from "food.com".

The statistics of our datasets are summarized in Table 4, where we can see the characters of these datasets vary a lot. For example, ML-1M is a smaller and denser dataset, but Amazon and Food are much larger and sparser. The domains of these datasets range from movie to e-commerce and food.

**Baselines.** Since the proposed method is a framework instead of a specific model, the effectiveness and generality should be demonstrated by applying it to different time-aware recommender models. In particular, the following representative models are leveraged to implement the base model $f$:

- **TimeSVD++** [24] is a well known matrix factorization model, where the time information is combined with the user and item embeddings, respectively.
- **BPTF** [42] is tensor factorization model, where the time information is modeled by an independent tensor dimension.
- **CoNCARS** [6] is deep time-aware recommender model, which leverages the CNN to capture the non-linear relationships among the users, items and time.

It should be noted that our paper focuses on time-aware recommendation, and the time-dependent methods, such as the sequential or session-based recommender models [15, 41] are beyond the scope of this paper, and thus not compared.

From the framework perspective, we introduce a baseline called **DANCER** [16], which, to our knowledge, is the only time-aware debiased framework. We predict the preference score by TMF [16], which can achieve better performance on our datasets. In addition, we also compare our framework with two static baselines: **SVD++**[22] is a widely-used matrix factorization model, where the rating is estimated by the inner product between the user and item latent vectors. **RD-DR**[9] is a recently proposed debiased recommender model, where the unmeasured confounders are captured by sensitivity analysis.

**Implementation details.** In the domain of time-aware recommendation, the time information is usually projected into IDs [3]. In general, there are two strategies: the first one (**S1**) is segmenting the complete time range of the dataset into many bins, and the indexes of the bins are regarded as the time IDs [16, 21]; The second one (**S2**) is indicating the time IDs as the indexes of the hours of a day or the days of a week [1, 5, 27]. We conduct our experiments based on both settings, where for the first strategy, we split the time range into seven bins and for the second strategy, we produce the IDs according to the days of a week.

Evaluating debiased recommender frameworks needs to build unbiased testing sets, in our experiments, we follow the common practice [29, 30, 40, 47] to resample the original dataset according to the sample observational frequencies. In particular, we use 50% interactions of each user for biased model training. To build the validation and testing sets, we sample from the remaining interactions. For a sample $(u, v, t, r)$, we select it according to the probabilities in proportional to $\frac{\max_{v', t'} O_{v't'}}{O_{vt}}$, where $O_{vt}$ is the number of times that $(v, t)$ appears in the dataset. The ratio between the validation and test sets is set as 1:1.

We evaluate the performance of our framework based on the commonly used metrics including Root Mean Squared Error(RMSE) and Mean Absolute Error(MAE) [17]. The hyper-parameters are tuned based on grid search. In particular, the learning rate and hidden embedding size are determined in the ranges of $[0.1, 0.01, 0.001, 0.0001]$ and $[32, 64, 128]$, respectively. $\Gamma_{uv}$ and $\Gamma_{uvt}$ are both tuned in the range of $[1.1, 1.3, 1.5, 1.8, 2]$. We empirically set the batch size as 1024. For the baselines, we tune their parameters in the same ranges to our framework's. For more implementation details, we refer the readers to our project released at https://www.cdtr.github.io/.

### 5.2 Overall Performance

The overall comparison results are presented in Table 1, where we can see: by incorporating the time information, TimeSVD++, BPTF

---

[3]https://grouplens.org/datasets/movielens/1m/
[4]http://jmcauley.ucsd.edu/data/amazon/
[5]https://www.kaggle.com/datasets/shuyangli94/food-com-recipes-and-user-interactions

**Table 1: Overall comparison between our framework (*i.e.*, CDTR) and the baselines. For each base model, we use bold fonts to label the best performance of the evaluated models. * indicates that the improvement of our framework against the best baseline is significant under paired-t test with $p < 0.05$.**

| Dataset | | ML-1M-S1 | | ML-1M-S2 | | Amazon-S1 | | Amazon-S2 | | Food-S1 | | Food-S2 | |
|---|---|---|---|---|---|---|---|---|---|---|---|---|---|
| Metric | | RMSE | MAE | RMSE | MAE | RMSE | MAE | RMSE | MAE | RMSE | MAE | RMSE | MAE |
| SVD++ [22] | | 1.0156 | 0.7994 | 0.9462 | 0.7443 | 0.9623 | 0.7367 | 1.0165 | 0.7633 | 1.0830 | 0.7397 | 1.0882 | 0.7419 |
| RD-DR [9] | | 0.9931 | 0.8037 | 0.9583 | 0.7718 | 0.9957 | 0.7282 | 0.9963 | 0.7274 | 0.9969 | **0.6491** | 1.0022 | 0.6520 |
| TimeSVD++ [23] | | 0.8797 | 0.6884 | 0.9004 | 0.7071 | 0.9356 | 0.6992 | 0.8843 | 0.6508 | 1.0523 | 0.7117 | 1.0502 | 0.7350 |
| DANCER [16] | | 0.8698 | 0.6804 | 0.8832 | 0.6909 | 0.8974 | 0.6534 | 0.8806 | 0.6302 | 1.0340 | 0.6897 | 1.0316 | 0.6827 |
| CDTR | IPS | 0.8712 | 0.6812 | 0.8786 | 0.6861 | 0.9021 | 0.6577 | 0.8825 | 0.6416 | 1.0318 | 0.6699 | 1.0336 | 0.6888 |
| | IPS-UM | 0.8707 | 0.6818 | 0.8757 | 0.6840 | 0.8888 | **0.6501** | 0.8803 | 0.6442 | 1.0311 | 0.6683 | 1.0272 | 0.6678 |
| | DR | 0.8661 | 0.6770 | 0.8724 | 0.6816 | 0.8974 | 0.6615 | 0.8801 | 0.6407 | 1.0117 | 0.6879 | 1.0159 | 0.7167 |
| | DR-UM | **0.8650*** | **0.6765*** | **0.8696*** | **0.6797*** | **0.8813*** | 0.6672 | **0.8790*** | **0.6401*** | **0.9959*** | 0.6504 | **0.9995*** | **0.6459*** |
| BPTF [42] | | 0.8941 | 0.7003 | 0.9147 | 0.7154 | 1.0368 | 0.7751 | 1.0596 | 0.7908 | 1.0162 | 0.6424 | 1.0162 | 0.6294 |
| DANCER [16] | | 0.8815 | 0.6899 | 0.9209 | 0.7217 | 1.0166 | 0.7604 | 1.0441 | 0.7758 | 1.0122 | 0.6294 | 1.0188 | 0.6248 |
| CDTR | IPS | 0.8885 | 0.6960 | 0.9087 | 0.7112 | 1.0143 | 0.7610 | 1.0378 | 0.7837 | 1.0078 | 0.6288 | 1.0057 | 0.6205 |
| | IPS-UM | 0.8817 | 0.6900 | 0.9073 | 0.7116 | 1.0045 | 0.7554 | 1.0265 | 0.7644 | 1.0038 | **0.6193** | 1.0000 | 0.6104 |
| | DR | 0.8797 | 0.6864 | 0.8895 | **0.6945** | 0.9575 | 0.6816 | 1.0312 | 0.7618 | 0.9613 | 0.6967 | 0.9982 | 0.6059 |
| | DR-UM | **0.8764*** | **0.6856*** | **0.8887*** | 0.6946 | **0.9204*** | **0.6509*** | **0.9513*** | **0.6913*** | **0.9200*** | 0.6727 | **0.9980*** | **0.5993*** |
| CoNCARS [6] | | 0.9335 | 0.7402 | 0.9325 | 0.7387 | 0.9387 | 0.6422 | 0.9375 | 0.6809 | 1.027 | 0.6543 | 1.0320 | 0.6729 |
| DANCER [16] | | 0.9184 | 0.7253 | 0.9171 | 0.7267 | 0.9127 | 0.6679 | 0.9226 | 0.6524 | 1.009 | 0.6417 | 0.9950 | 0.6361 |
| CDTR | IPS | 0.9170 | 0.7250 | 0.9152 | 0.7223 | 0.9126 | 0.6646 | 0.9291 | 0.6849 | 1.0140 | 0.6237 | 0.9940 | 0.6083 |
| | IPS-UM | 0.9149 | 0.7224 | 0.9115 | 0.7235 | 0.9100 | **0.6384** | 0.9090 | 0.6765 | 0.9928 | 0.6290 | 0.9840 | 0.687 |
| | DR | 0.9044 | 0.7127 | 0.9043 | 0.7197 | 0.9030 | 0.6400 | 0.9076 | 0.6515 | 0.9766 | **0.5526** | 0.9916 | 0.6034 |
| | DR-UM | **0.8990*** | **0.7087*** | **0.8932*** | **0.7020*** | **0.8940*** | 0.6495 | **0.9061*** | **0.6294*** | **0.9749*** | 0.5690 | **0.9826*** | **0.5856*** |

**Table 2: Comparison between our framework and its variants. The best performance is labeled by bold fonts.**

| Method | ML1M | | Food | |
|---|---|---|---|---|
| Metric | RMSE | MAE | RMSE | MAE |
| TimeSVD++ | 0.9440 | 0.7457 | 1.0523 | 0.7117 |
| IPS | **0.8712** | **0.6812** | **1.0318** | **0.6699** |
| w/o V | 0.9055 | 0.7125 | 1.1086 | 0.7612 |
| w/o T | 0.9173 | 0.7231 | 1.1080 | 0.7595 |
| IPS-UM | **0.8707** | **0.6818** | **1.0311** | **0.6683** |
| w/o V | 0.9651 | 0.7510 | 1.1089 | 0.7616 |
| w/o T | 0.9652 | 0.7511 | 1.1088 | 0.7614 |
| DR | **0.8661** | **0.6770** | **1.0117** | **0.6879** |
| w/o V | 0.8727 | 0.6799 | 1.1370 | 0.8294 |
| w/o T | 0.8860 | 0.6939 | 1.0904 | 0.7697 |
| DR-UM | **0.8650** | **0.6765** | **0.9959** | **0.6504** |
| w/o V | 0.8729 | 0.6798 | 1.0091 | 0.6954 |
| w/o T | 0.8730 | 0.6798 | 1.0089 | 0.6919 |

and CoNCARS can usually achieve better performance than the static models like SVD++. This is as expected, which demonstrates the effectiveness of the time information for estimating the user preference. For each base model, DANCER can improve the performance across different metrics and datasets, and the results are consistent in most cases. The observation agrees with the previous work [16]. The reason can be that, in DANCER, the training samples are reweighted according to the observational frequencies. The learned model can equally treat different items and times, which is more aligned with the unbiased test sets. However, in the base model, the sample weights are not adjusted, thus the learned model may put more focus on the advantaged items/times, which may not perform well when different items/times are equally evaluated.

Encouragingly, our framework consistently outperforms baselines across different datasets, evaluation metrics, and base models in most cases. On average, our framework can improve the performance of the best baseline up to 4.73% and 6.23% on RMSE and MAE, respectively. Comparing with DANCER, we find that the basic IPS method of our framework does not exhibit significant superiority.

This is not surprising, since they have similar major components, and are only different in how to estimate the IPS. By extending the IPS method to the doubly robust model, our framework can usually achieve better performances. We speculate that the introduction of the imputation model reduces the variance of our framework, and more easily obtain the unbiased estimator, which is aligned with the test set.

Modeling the unmeasured confounders is effective, which is evidenced by the improved performances of IPS-UM and DR-UM as compared with IPS and DR, respectively. By combining the doubly robust method and the component for capturing the unmeasured confounders, the performances are further improved, which suggests that these modules are both important, complementary to each other.

## 5.3 Ablation Studies

In the above section, we have evaluated the effectiveness of our framework based on four implementations, that is, IPS, DR, IPS-UM and DR-UM. In this section, we would like to study whether the item- and time-level IPS's in these implementations are both necessary. To answer this question, we compare our framework with its two variants: X(w/o V), we remove the item-level IPS, where $X$ indicates an implementation of our framework. In X(w/o T), we remove the time-level IPS. We base the experiment on ML-1M and Food, and use TimeSVD++ as the base model. We use S1 to process the time information (i.e., discretizing the time to days in a week).

From the results presented in Table 2, we can see: the winner between X(w/o V) and X(w/o T) varies on different datasets and implementations of our framework, but the performance gains are not significant. In some cases, X(w/o V) and X(w/o T) may even perform worse than the base model. For any implementation, combining the item- and time-level IPS can always achieve the best performance, and the results are consistent on different datasets. These observations suggest that simultaneously correcting the item-

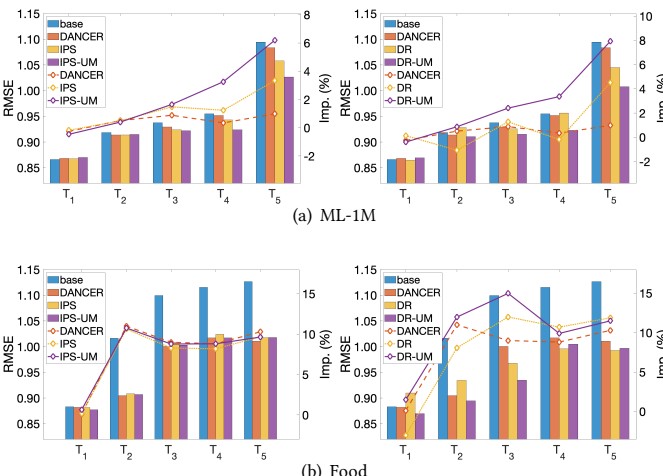

**Figure 3: Debiasing effects of our framework on the items and times with different observational frequencies.** *Imp.* **refers to the improvement over the baseline. We sort the samples based on the inverse of item-time occurrence frequency.** $\{T1, T2, T3, T4, T5\}$ **represent the samples in the intervals of** $[0, 20\%]$**,** $[20, 40\%]$**,** $[40, 60\%]$**,** $[60, 80\%]$**, and** $[80, 100\%]$ **according to their occurrence frequency.**

and time-level biases is necessary, and only one type of IPS is usually sub-optimal.

### 5.4 Fine-grained Debiasing Effects

In this section, we study the debiasing effect of our framework for the samples with different observational frequencies. To achieve this goal, we firstly sort the item-time pairs according to their frequencies in the training set, where a higher ranking corresponds a more popular item-time pair. Then we divide the testing set into five subsets $\{T1, T2, T3, T4, T5\}$, which only contain the samples with the item-time pairs ranked between $[0, 20\%]$, $[20, 40\%]$, $[40, 60\%]$, $[60, 80\%]$ and $[80, 100\%]$, respectively. In the experiments, we keep the training set the same as before, and report the performances on T1, T2, T3, T4 and T5, respectively. The other experiment settings follow the above section.

From the results presented in Figure 3, we can see: on the testing sets with popular samples (*e.g.*, T1), our framework does not outperform the base model, and sometimes, the performance is actually degraded. This is not surprising, since in the training process of our framework, the popular samples are imposed with smaller weights, which are valued less than the base model. On the testing sets with non-popular samples (*e.g.*, T4 and T5), our framework can achieve much better performance, which agrees with our expectation. This observation suggests that the debiasing effect of our framework are more effective for the samples with lower observational frequencies. By changing the testing sets from T1 to T5, we find that the performance improvement of our framework is continually enlarged, which is encouraging and demonstrates the potential of our framework for handling the extremely biased samples.

**Table 3: Performance of our framework with different bias severity. The best performance is labeled by bold fonts.**

| Dataset | | ML-1M | | | | | |
|---|---|---|---|---|---|---|---|
| $\rho$ | | 0.5 | | 1 | | 2 | |
| Metric | | RMSE | MAE | RMSE | MAE | RMSE | MAE |
| TimeSVD++ | | 0.9622 | 0.7535 | 1.0976 | 0.8419 | 1.440 | 1.0842 |
| CDTR | IPS | 0.9516 | 0.7491 | 1.0425 | 0.8122 | 1.3122 | 1.0086 |
| | IPS-UM | 0.9530 | **0.7489** | 1.0042 | **0.7896** | 1.2266 | 0.9508 |
| | DR | 0.9794 | 0.7810 | 1.0415 | 0.8235 | 1.2832 | 0.9967 |
| | DR-UM | **0.9510** | 0.7569 | **1.0035** | 0.8008 | **1.2001** | **0.9454** |
| Dataset | | Food | | | | | |
| $\rho$ | | 0.5 | | 1 | | 2 | |
| Metric | | RMSE | MAE | RMSE | MAE | RMSE | MAE |
| TimeSVD++ | | 1.0385 | 0.6982 | 1.0418 | 0.7007 | 1.0453 | 0.6944 |
| CDTR | IPS | 1.0215 | 0.6727 | 1.0243 | 0.6732 | 1.0266 | 0.6746 |
| | IPS-UM | 1.0195 | 0.6666 | 1.0240 | 0.6672 | 1.0249 | 0.6880 |
| | DR | 1.0119 | 0.6632 | 1.0098 | 0.6596 | 1.0213 | 0.6763 |
| | DR-UM | **1.0109** | **0.6379** | **1.0048** | **0.6466** | **1.0093** | **0.6434** |

### 5.5 Influence of the Bias Severity

In this section, we would like to study whether our framework can work well for differently biased training sets. To study this problem, we simulate different bias severities by resampling the original training set. In particular, we sample the dataset according to the frequencies of the item-time pairs, that is, $[\frac{O_{vt}}{\max_{v', t'} O_{v't'}}]^{\rho}$, where $O_{vt}$ is the number of times that $(v, t)$ is observed in the training set. By setting a larger $\rho$, the dataset is more biased, since the popular samples are more likely to be selected, and the unpopular ones have smaller chances to be kept in the training set. In the experiments, we tune $\rho$ in the range of $[0.5, 1, 2]$, and the other settings follow the above section.

From the results presented in Table 3, we can see: for each $\rho$, the best of our framework can always achieve better performance than the base model, and the results are consistent on both datasets. These results manifest that the effectiveness of our framework is not influenced by the bias severity. In most cases, DR-UM can outperform the other implementations of our framework, which agrees with the observations in Table 1, and further demonstrates the effectiveness of the doubly robust and sensitivity analysis modules. Interestingly, as the bias becomes severer, the performance improvement of our framework is enlarged. On average, the performance improvements of our framework are 1.91%, 6.06% and 10.05% when $\rho = 0.5$, $\rho = 1$ and $\rho = 2$, respectively. This observation demonstrates the potential of our framework for the application scenarios where the datasets are extremely biased.

### 6 CONCLUSIONS AND FUTURE WORK

In this paper, we propose a causally debiased recommender framework, where we firstly design a basic IPS model and then extend it to the doubly robust method to further improve it performance. In addition, we also propose a sensitivity analysis method to capture the unmeasured confounders. Beyond introducing the model designs, we also present a series of theoretical analysis, which is expected to provide more in depth understandings on our framework. Actually, there is still much room left for improvement. To begin with, one can extend our framework to the settings of sequential recommendation, where the time is not used as a context information, but leveraged to chronologically sort the items. In addition, it could be also interesting to directly model the continuous time information, where the propensity score should follow a continuous distribution.

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

# A DATASET DETAILS

**Table 4: Statistics of the datasets.**

| Dataset | Users | Items | Interactions | Sparsity |
|---|---|---|---|---|
| ML-1M | 6,041 | 3,707 | 1,000,209 | 95.53% |
| Amazon | 22,879 | 115,083 | 238,692 | 99.99% |
| Food | 23,087 | 211,040 | 872,021 | 99.98% |

# B PROOF OF THEOREMS

## B.1 Proof of Theorem 1

PROOF. For clear presentation, we denote $p_O^V(v|u)$ and $p_O^T(t|u,v)$ by $p_O^V$ and $p_O^T$, respectively. Then, we have:

$$E\left[L_{\text{IPS}}\right]$$

$$=E\left[\frac{1}{|\mathcal{U}|}\sum_{u\in\mathcal{U}}\left\{\frac{1}{|\mathcal{V}_u|}\sum_{v\in\mathcal{V}_u}\frac{p^V(v)}{p_O^V}\left[\frac{1}{|\mathcal{T}_{uv}|}\sum_{t\in\mathcal{T}_{uv}}\frac{p^T(t)}{p_O^T}\delta_{uvt}\right]\right\}\right]$$

$$=\frac{1}{|\mathcal{U}|}\sum_{u\in\mathcal{U}}\left\{\frac{1}{|\mathcal{V}_u|}\sum_{v\in\mathcal{V}_u}E_{v\sim p_O^V}\left[\frac{p^V(v)}{p_O^V}\frac{1}{|\mathcal{T}_{uv}|}\sum_{t\in\mathcal{T}_{uv}}E_{t\sim p_O^T}\left[\frac{p^T(t)}{p_O^T}\delta_{uvt}\right]\right]\right\}$$

$$=\frac{1}{|\mathcal{U}|}\sum_{u\in\mathcal{U}}\left\{\frac{1}{|\mathcal{V}_u|}\sum_{v\in\mathcal{V}_u}E_{v\sim p_O^V}\left[\frac{p^V(v)}{p_O^V}\frac{1}{|\mathcal{T}_{uv}|}\sum_{t\in\mathcal{T}_{uv}}E_{t\sim p^T}\left[\delta_{uvt}\right]\right]\right\}$$

$$=\frac{1}{|\mathcal{U}|}\sum_{u\in\mathcal{U}}\left\{\frac{1}{|\mathcal{V}_u|}\sum_{v\in\mathcal{V}_u}E_{v\sim p_O^V}\left[\frac{p^V(v)}{p_O^V}E_{t\sim p^T}\left[\delta_{uvt}\right]\right]\right\}$$

$$=\frac{1}{|\mathcal{U}|}\sum_{u\in\mathcal{U}}\left\{\frac{1}{|\mathcal{V}_u|}\sum_{v\in\mathcal{V}_u}E_{v\sim p^V}\left[E_{t\sim p^T}\left[\delta_{uvt}\right]\right]\right\}$$

$$=\frac{1}{|\mathcal{U}|}\sum_{u\in\mathcal{U}}E_{v\sim p^V}\left[E_{t\sim p^T}\left[\delta_{uvt}\right]\right]$$

$$=L_{\text{ideal}}$$

## B.2 Proof of Theorem 2

PROOF. Suppose $\hat{p}_O^V$ and $\hat{p}_O^T$ are short for $\hat{p}_O^V(v|u)$ and $\hat{p}_O^T(t|u,v)$, respectively. Then, we have:

$$E[L_{\text{DR}}]=\frac{1}{|\mathcal{U}|}\sum_{u\in\mathcal{U}}\left\{\frac{1}{|\mathcal{V}_u|}\sum_{v\in\mathcal{V}_u}E_{p_O^V}\left[\frac{p^V(v)}{\hat{p}_O^V}E_{p_O^T}\left[\frac{p^T(t)}{\hat{p}_O^T}\triangle_{uvt}\right]\right]\right\}$$

$$+E\left[\frac{1}{|\mathcal{U}||\mathcal{V}||\mathcal{T}|}\sum_{u\in\mathcal{U}}\sum_{v'\in\mathcal{V}}\sum_{t'\in\mathcal{T}}\hat{\delta}_{uv't'}\right]$$

$$=\frac{1}{|\mathcal{U}||\mathcal{V}||\mathcal{T}|}\sum_{u\in\mathcal{U}}\sum_{v\in\mathcal{V}}\sum_{t\in\mathcal{T}}\left[\triangle_{uvt}\frac{p_O^V p_O^T}{\hat{p}_O^V\hat{p}_O^T}+\hat{\delta}_{uvt}\right]$$

$$=\frac{1}{|\mathcal{U}||\mathcal{V}||\mathcal{T}|}\sum_{u\in\mathcal{U}}\sum_{v\in\mathcal{V}}\sum_{t\in\mathcal{T}}\left[\triangle_{uvt}\frac{s_{uvt}}{\hat{s}_{uvt}}+\hat{\delta}_{uvt}\right],$$

where the second equation holds because the expectation is taken over $v$ and $t$, and irrelevant with $v'$ and $t'$. Then, by bringing

$s_{uvt}=p_O^V(v|u)p_O^T(t|u,v)$ and $\hat{s}_{uvt}=\hat{p}_O^V(v|u)\hat{p}_O^T(t|u,v)$ into the above equation, we have:

$$Bias(L_{\text{DR}})=|E\left[L_{\text{DR}}\right]-L_{ideal}|$$

$$=\left|\frac{1}{|\mathcal{U}||\mathcal{V}||\mathcal{T}|}\sum_{u\in\mathcal{U}}\sum_{v\in\mathcal{V}}\sum_{t\in\mathcal{T}}\left[\triangle_{uvt}\frac{s_{uvt}}{\hat{s}_{uvt}}+\hat{\delta}_{uvt}-\delta_{uvt}\right]\right|$$

$$=\left|\frac{1}{|\mathcal{U}||\mathcal{V}||\mathcal{T}|}\sum_{u\in\mathcal{U}}\sum_{v\in\mathcal{V}}\sum_{t\in\mathcal{T}}\triangle_{uvt}(\frac{s_{uvt}}{\hat{s}_{uvt}}-1)\right| \quad (4)$$

$$=\left|\frac{1}{|\mathcal{U}||\mathcal{V}||\mathcal{T}|}\sum_{u\in\mathcal{U}}\sum_{v\in\mathcal{V}}\sum_{t\in\mathcal{T}}\triangle_{uvt}\frac{s_{uvt}-\hat{s}_{uvt}}{\hat{s}_{uvt}}\right|.$$

## B.3 Proof of Theorem 3

PROOF. The variance of $L_{\text{IPS}}$ is:

$$\text{Var}(L_{\text{IPS}})=\frac{1}{|\mathcal{U}|^2}\sum_{u\in\mathcal{U}}\sum_{v\in\mathcal{V}_u}\sum_{t\in\mathcal{T}_{uv}}\text{Var}(X_{uvt}).$$

To compare $\text{Var}(L_{\text{DR}})$ and $\text{Var}(L_{\text{IPS}})$, we have:

$$\text{Var}(L_{\text{DR}})-\text{Var}(L_{\text{IPS}})=\frac{1}{|\mathcal{U}|^2}\sum_{u\in\mathcal{U}}\sum_{v\in\mathcal{V}_u}\sum_{t\in\mathcal{T}_{uv}}\text{Cov}(Y_{uvt}-2X_{uvt},Y_{uvt}).$$

Remember that by objective $L_{\text{imp}}$, we actually hope that $X_{uvt}=Y_{uvt}$. In such a scenario, $\text{Cov}(Y_{uvt}-2X_{uvt},Y_{uvt})=\text{Cov}(-Y_{uvt},Y_{uvt})<0$, which means $\text{Var}(L_{\text{DR}})<\text{Var}(L_{\text{IPS}})$.

## B.4 Proof of Theorem 4

PROOF. To begin with, we define

$$L'_{\text{IPS-UM}}=\frac{1}{|\mathcal{U}|}\sum_{u\in\mathcal{U}}l_u,$$

where

$$l_u=\frac{1}{|\mathcal{V}_u|}\sum_{v\in\mathcal{V}_u}\frac{1}{|\mathcal{T}_{uv}|}\sum_{t\in\mathcal{T}_{uv}}\frac{p^V(v)p^T(t)}{\overline{p}_{uv}\overline{p}_{uvt}}\delta_{uvt}.$$

Let $\hat{p}_{uv}$ and $\hat{p}_{uvt}$ be the initially estimated propensity scores, which do not consider the unmeasured confounders, then we have the following lemma on the uncertainty sets:

LEMMA B.1. *The constraints $\overline{p}_{uv}\in A_{uv}$ and $\overline{p}_{uvt}\in B_{uvt}$ are equal to*

$$|\log(\frac{\hat{p}_{uv}}{1-\hat{p}_{uv}})-\log(\frac{\overline{p}_{uv}}{(1-\overline{p}_{uv})})|\le b-a$$

*and*

$$|\log(\frac{\hat{p}_{uvt}}{1-\hat{p}_{uvt}})-\log(\frac{\overline{p}_{uvt}}{(1-\overline{p}_{uvt})})|\le d-c.$$

Based on Lemma B.1, we rewrite $L_{\text{IPS-UM}}$ in the form of

$$L_{\text{IPS-UM}}$$

$$=\max_{\overline{p}_{uv},\overline{p}_{uvt}}L'_{\text{IPS-UM}}-\sum_{u\in\mathcal{U}}\sum_{v\in\mathcal{V}}\sum_{t\in\mathcal{T}}\lambda_1|\log(\frac{\hat{p}_{uv}}{1-\hat{p}_{uv}})-\log(\frac{\overline{p}_{uv}}{(1-\overline{p}_{uv})})|$$

$$-\sum_{u\in\mathcal{U}}\sum_{v\in\mathcal{V}}\sum_{t\in\mathcal{T}}\lambda_2|\log(\frac{\hat{p}_{uvt}}{1-\hat{p}_{uvt}})-\log(\frac{\overline{p}_{uvt}}{(1-\overline{p}_{uvt})})|,$$

where we replace the constraints of $\overline{p}_{uv} \in A_{uv}$ and $\overline{p}_{uvt} \in B_{uvt}$ by adding regularizers in the objective; $\lambda_1$ and $\lambda_2$ are regularizer coefficients. Then we have the following:

Suppose $p_{uv}^*$ and $p_{uvt}^*$ are the real propensity scores, then we have:

$$
\begin{aligned}
&L_{\text{ideal}} - E_{p_{uv}^*, \, p_{uvt}^*}[L'_{\text{IPS-UM}}] \\
&= \frac{1}{|\mathcal{U}||\mathcal{V}||\mathcal{T}|} \sum_{u \in \mathcal{U}} \sum_{v \in \mathcal{V}} \sum_{t \in \mathcal{T}} (1 - \frac{p_{uv}^* p_{uvt}^*}{\overline{p}_{uv} \overline{p}_{uvt}}) \delta_{uvt}.
\end{aligned} \quad (5)
$$

According to the Hoeffding's inequality, and assuming $l_u \in [c_u, d_u]$, $|c_u - d_u| \leq B$, then the expectation of $L'_{\text{IPS-UM}}$ is bounded by the following value with probability at least $1 - \eta$:

$$
E_{p_{uv}^*, \, p_{uvt}^*}[L'_{\text{IPS-UM}}] \leq L'_{\text{IPS-UM}} + B\sqrt{\frac{1}{2|\mathcal{U}|} log(\frac{2|\mathcal{H}|}{\eta})}.
$$

Let

$$
C = B\sqrt{\frac{1}{2|\mathcal{U}|} log(\frac{2|\mathcal{H}|}{\eta})}, \alpha_{uvt} = \frac{\delta_{uvt}}{|\mathcal{U}||\mathcal{V}||\mathcal{T}|\overline{p}_{uv}\overline{p}_{uvt}}.
$$

Then,

$$
\begin{aligned}
&L_{\text{ideal}} = L_{\text{ideal}} - E_{p_{uv}^*, \, p_{uvt}^*}[L'_{\text{IPS-UM}}] + E_{p_{uv}^*, \, p_{uvt}^*}[L'_{\text{IPS-UM}}] \\
&\leq \frac{1}{|\mathcal{U}||\mathcal{V}||\mathcal{T}|} \sum_{u \in \mathcal{U}} \sum_{v \in \mathcal{V}} \sum_{t \in \mathcal{T}} (1 - \frac{p_{uv}^* p_{uvt}^*}{\overline{p}_{uv} \overline{p}_{uvt}}) \delta_{uvt} + L'_{\text{IPS-UM}} + C \\
&\leq \sum_{u \in \mathcal{U}} \sum_{v \in \mathcal{V}} \sum_{t \in \mathcal{T}} \alpha_{uvt}(\overline{p}_{uv}\overline{p}_{uvt} - p_{uv}^* p_{uvt}^*) + L'_{\text{IPS-UM}} + C \\
&\leq \sum_{u \in \mathcal{U}} \sum_{v \in \mathcal{V}} \sum_{t \in \mathcal{T}} [\alpha_{uvt}(\overline{p}_{uv}\overline{p}_{uvt} - \hat{p}_{uv}\hat{p}_{uvt}) \\
&\quad + \alpha_{uvt}(\hat{p}_{uv}\hat{p}_{uvt} - p_{uv}^* p_{uvt}^*)] + L'_{\text{IPS-UM}} + C \\
&\leq \sum_{u \in \mathcal{U}} \sum_{v \in \mathcal{V}} \sum_{t \in \mathcal{T}} \alpha_{uvt}(\overline{p}_{uv}\overline{p}_{uvt} - \hat{p}_{uv}\hat{p}_{uvt}) + L'_{\text{IPS-UM}} + C_1 \\
&= \sum_{u \in \mathcal{U}} \sum_{v \in \mathcal{V}} \sum_{t \in \mathcal{T}} \alpha_{uvt}(\overline{p}_{uv}\overline{p}_{uvt} - \overline{p}_{uv}\hat{p}_{uvt} + \overline{p}_{uv}\hat{p}_{uvt} - \hat{p}_{uv}\hat{p}_{uvt}) \\
&\quad + L'_{\text{IPS-UM}} + C_1 \\
&= \sum_{u \in \mathcal{U}} \sum_{v \in \mathcal{V}} \sum_{t \in \mathcal{T}} [\alpha_{uvt}\overline{p}_{uv}(\overline{p}_{uvt} - \hat{p}_{uvt}) + \alpha_{uvt}\hat{p}_{uvt}(\overline{p}_{uv} - \hat{p}_{uv})] \\
&\quad + L'_{\text{IPS-UM}} + C_1 \\
&\leq \sum_{u \in \mathcal{U}} \sum_{v \in \mathcal{V}} \sum_{t \in \mathcal{T}} [\alpha_{uvt}\overline{p}_{uv}(2s_2 - \overline{p}_{uvt} - \hat{p}_{uvt}) \\
&\quad + \alpha_{uvt}\hat{p}_{uvt}(2s_2 - \overline{p}_{uv} - \hat{p}_{uv})] + L'_{\text{IPS-UM}} + C_1 \\
&\leq \frac{\kappa s_2}{|\mathcal{U}||\mathcal{V}||\mathcal{T}|s_1^2} \sum_{u \in \mathcal{U}} \sum_{v \in \mathcal{V}} \sum_{t \in \mathcal{T}} [(2s_2 - |\overline{p}_{uvt}| - |\hat{p}_{uvt}|) \\
&\quad + (2s_2 - |\overline{p}_{uv}| - |\hat{p}_{uv}|)] + L'_{\text{IPS-UM}} + C_1 \\
&\leq \frac{\kappa s_2}{|\mathcal{U}||\mathcal{V}||\mathcal{T}|s_1^2} \sum_{u \in \mathcal{U}} \sum_{v \in \mathcal{V}} \sum_{t \in \mathcal{T}} [(2s_2 - |\overline{p}_{uvt} - \hat{p}_{uvt}|) \\
&\quad + (2s_2 - |\overline{p}_{uv} - \hat{p}_{uv}|)] + L'_{\text{IPS-UM}} + C_1 \\
&\leq -\frac{\kappa s_2}{|\mathcal{U}||\mathcal{V}||\mathcal{T}|s_1^2} \sum_{u \in \mathcal{U}} \sum_{v \in \mathcal{V}} \sum_{t \in \mathcal{T}} (|\overline{p}_{uvt} - \hat{p}_{uvt}| + |\overline{p}_{uv} - \hat{p}_{uv}|) \\
&\quad + L'_{\text{IPS-UM}} + C_2,
\end{aligned} \quad (6)
$$

where

$$
C_1 = \kappa(\frac{s_2^2}{s_1^2} - 1) + C, C_2 = \frac{4\kappa s_2^2}{s_1^2} + C_1.
$$

LEMMA B.2. *Suppose $f(x) = \log\frac{x}{1-x}$ and $x \in [s_1, s_2]$, then*

$$
|f(x) - f(y)| \leq \eta|x - y|
$$

*, where $\eta = \frac{1}{s_1} + \frac{1}{1-s_2}$.*

PROOF. According to the Taylor expansion, we have $\log x = \sum_{n=1}^{\infty}(-1)^{n-1}\frac{(x-1)^n}{n}$ and $\log(1-x) = -\sum_{n=1}^{\infty}\frac{x^n}{n}$. Thus,

$$
\begin{aligned}
&|\log\frac{x}{1-x} - \log\frac{y}{1-y}| = |\log x - \log y - [\log(1-x) - \log(1-y)]| \\
&= |\sum_{n=1}^{\infty}(-1)^{n-1}\frac{(x-1)^n}{n} - \sum_{n=1}^{\infty}(-1)^{n-1}\frac{(y-1)^n}{n} \\
&\quad + \sum_{n=1}^{\infty}\frac{x^n}{n} - \sum_{n=1}^{\infty}\frac{y^n}{n}| \\
&= |\sum_{n=1}^{\infty}(-1)^{n-1}\frac{(x-1)^n - (y-1)^n}{n} + \sum_{n=1}^{\infty}\frac{x^n - y^n}{n}| \\
&= |\sum_{n=1}^{\infty}(-1)^{n-1}\frac{(x-y)[(x-1)^{n-1} + (x-1)^{n-2}(y-1)...+(y-1)^{n-1}]}{n} \\
&\quad + \sum_{n=1}^{\infty}\frac{(x-y)[x^{n-1} + x^{n-2}y... + y^{n-1}]}{n}| \\
&= |x-y||\sum_{n=1}^{\infty}\frac{[(1-x)^{n-1} + (1-x)^{n-2}(1-y)... + (1-y)^{n-1}]}{n} \\
&\quad + \sum_{n=1}^{\infty}\frac{[x^{n-1} + x^{n-2}y... + y^{n-1}]}{n}| \\
&\leq |x-y||\sum_{n=1}^{\infty}\frac{n(1-s_1)^{n-1}}{n} + \sum_{n=1}^{\infty}\frac{ns_2^{n-1}}{n}| \\
&= |x-y||\sum_{n=1}^{\infty}(1-s_1)^{n-1} + \sum_{n=1}^{\infty}s_2^{n-1}| \\
&\leq (\frac{1}{s_1} + \frac{1}{1-s_2})|x-y| = \eta|x-y|.
\end{aligned}
$$

Let $\lambda = \frac{\kappa s_2}{|\mathcal{U}||\mathcal{V}||\mathcal{T}|s_1^2}$. Then based on Lemma B.2, we have:

$$
\begin{aligned}
L_{\text{ideal}} &\leq -\lambda \sum_{u \in \mathcal{U}} \sum_{v \in \mathcal{V}} \sum_{t \in \mathcal{T}} (|\overline{p}_{uvt} - \hat{p}_{uvt}| + |\overline{p}_{uv} - \hat{p}_{uv}|) \\
&\quad + L'_{\text{IPS-UM}} + C_2 \\
&\leq -\lambda \sum_{u \in \mathcal{U}} \sum_{v \in \mathcal{V}} \sum_{t \in \mathcal{T}} \frac{1}{\eta}|\log(\frac{\hat{p}_{uv}}{1-\hat{p}_{uv}}) - \log(\frac{\overline{p}_{uv}}{(1-\overline{p}_{uv})})| \\
&\quad -\lambda \sum_{u \in \mathcal{U}} \sum_{v \in \mathcal{V}} \sum_{t \in \mathcal{T}} \frac{1}{\eta}|\log(\frac{\hat{p}_{uvt}}{1-\hat{p}_{uvt}}) - \log(\frac{\overline{p}_{uvt}}{(1-\overline{p}_{uvt})})| \\
&\quad + L'_{\text{IPS-UM}} + C_2 \\
&\leq L_{\text{IPS-UM}} + Const.
\end{aligned}
$$

$\square$