# OpenReview forum: "Causally Debiased Time-aware Recommendation"
_ACM.org/TheWebConf/2024/Conference — TheWebConf24_

### Official Review · Reviewer_NPL9 · 2023-11-16

**Novelty:** 3
**Technical Quality:** 3

**Review:**

1. Causal Graph
- I cannot fully buy the Figure 2a. why T is affected by U and V?
- Instead, I think the time T changes the user and item (i.e., the user and item representation is affected by the time).
- we get the affected U' and I' with the graph U -> U' <- T -> I' <- I  (U, I are the general user and item variables, and U' and I' are time-aware variables.)
- and the rating R is affected by U' and I', with U' -> R <- I'
- Also, I cannot understand why Figure 2b is an 'ideal' learning objective. To accurately estimate the user feedback on all items at all times, the model needs to utilize the item information and the time information.
- You cannot call Figure 2 a causal graph, because it's your desired data generation process, not the real world.
- After all, the authors do not utilize the causal graph. Why do we need this? If you really want to utilize the causal graph, you need to adopt any causality-aware techniques like total effect or reference values.

2. Method
- On the other hand, the proposed method itself is sound.
- If someone wants to deal with the bias of the observation and the time, she can adopt IPS on both.
- However, I personally recommend using the observation variable for the item and the time. p(v|u), p(t|u,v) do not make any probabilistic sence.
- Instead, how about $p(o_{u,v}=1|u)$ and $p(o_{u,v,t}=1|u,v)$?
- How do you estimate the propensity scores? In section 4.4, I can see the modeling for the propensity score, but cannot find any loss functions for that.

3. Contribution
- The contribution of this work is somewhat limited.
- They adopt the existing DR estimator both for the item observation and interaction time.
- They adopt the existing Sensitivity Analysis for the hidden confounders.

4. Experiment
- Why do not compare 'Learning heterogeneous temporal patterns of user preference for timely recommendation, WWW 2021'?
- If this method is adopted in the real-world system, do we have to make inferences for all users and all items in every time slot?

**Questions:**

Please refer to Review.

**Reviewer Confidence:**

4: The reviewer is certain that the evaluation is correct and very familiar with the relevant literature

**Scope:**

3: The work is somewhat relevant to the Web and to the track, and is of narrow interest to a sub-community

---

### Official Review · Reviewer_cQaE · 2023-11-21

**Novelty:** 5
**Technical Quality:** 5

**Review:**

In this submission, the authors formulate the task of time-aware recommendation by a causal graph, by analyzing the causes of the item- and time-level biases, and adjust the training samples based on the inverse propensity score (IPS) to correct these biases.

Pros:
1) The paper provides a detailed theoretical analysis of the proposed models.
2) Extensive experiments are conducted, demonstrating the effectiveness of the framework.
3) The paper explores an intriguing and significant area in recommendation system.

Cons:
1) The improvement over the strongest existing baseline is relatively modest.
2) The evaluation focuses solely on RMSE and MAE. Incorporating additional metrics like Recall, F1, and NDCG could provide a more comprehensive assessment.
3) In Section 5.4, the scenarios T1, T2, ..., T5 seem impractical for real-world recommendation contexts. Typically, only the top 20 (or fewer) items are presented to users, which might undermine the effectiveness of the debiasing method.

**Questions:**

Are there any other observations related to the dataset segmentation methods, S1 and S2? For example, changing the number of bins or the use the days of a year.

What do the distributions of the time information of user-item interactions for all three datasets look like based on your segmentation?

**Reviewer Confidence:**

3: The reviewer is confident but not certain that the evaluation is correct

**Scope:**

3: The work is somewhat relevant to the Web and to the track, and is of narrow interest to a sub-community

---

### Official Review · Reviewer_eeDQ · 2023-11-21

**Novelty:** 4
**Technical Quality:** 5

**Review:**

### Summary

This work studies on dynamic debiased recommendation, and proposes a new method based on doubly-robust strategy. This work has the following strengths and weaknesses:


### Strengths

1. This work studies on an important problem.
2. This work proposes a novel doubly robust method to tackle both popularity bias and temporal bias.
3. Both theoretical analyses and empirical experiments are conducted to validate the effectiveness of the proposal.

### Weaknesses

1.	Although this work introduces a novel debiasing recommendation method utilizing a doubly robust approach, the majority of the techniques appear to be direct extensions of existing methods such as temporal Inverse Propensity Scoring (IPS) [16] and sensitivity analysis [9]. Consequently, the novelty is incremental.

2.	The experiments have some limitations:

a)	The baselines are weak. Some SOTA debiaisng strategies should be considered such as [a1][a2]. Notably, [a1] has demonstrated effective mitigation of temporal bias and warrants consideration.
b)	Ranking metrics like NDCG, Precision and Recall should be included, which are more closely align with the recommendation objective.
c)	It would be better to include more advanced backbone model, especially those sequential recommendation models, which draw mch attention from the RS community.

3.	This work misses some important related work on recommendation debiasing including:

a)	Some work on addressing popularity bias:

[a1] SIGIR’23: Invariant Collaborative Filtering to Popularity Distribution Shift
[a2] CIKM’22: Countering Popularity Bias by Regularizing Score Difference
[a3] TKDE’22：Popularity bias is not always evil: Disentangling benign and harmful bias for recommendation

b)	Some work on DR-based recommendation debiasing:

[b1] SIGIR’21: AutoDebias: Learning to debias for recommendation
[b2] ICML’23: StableDR: Stabilized Doubly Robust Learning for Recommendation on Data Missing Not at Random
[b3] CIKM’23: CDR: Conservative Doubly Robust Learning for Debiased Recommendation

### Overall evaluation

In summary, while this work has some limitations, I appreciate this work studies on an important problem and deliver a theoretical-sound method. I think this work is borderline but I relatively incline to weak accept if the authors could address my concerns.

**Questions:**

Please refer to weaknesses.

**Reviewer Confidence:**

4: The reviewer is certain that the evaluation is correct and very familiar with the relevant literature

**Scope:**

4: The work is relevant to the Web and to the track, and is of broad interest to the community

---

### Official Review · Reviewer_moET · 2023-11-23

**Novelty:** 6
**Technical Quality:** 6

**Review:**

Summary
In this paper, the authors aim to solve the problem of dynamic debiased recommendation. To achieve this goal, the authors proposed an IPS based model, causally debiased time-aware recommender framework, which is from the casual perspective. And the authors designed an IPS method was designed to correct the item- and time-level biases, and then extend the IPS method to the doubly robust model.  Besides, a sensitivity is applied to capture the unmeasured confounders. In the experiment, the authors evaluate the performance over the real-world dataset comparing to several baselines.
Pros
Originality
It is novel and interesting that the authors proposed a debiased time-aware recommender framework from the casual perspective, which considers the user preference. And it is the first time that the authors draw a connection between the proposed method and the ideal learning objective.
Quality
From the perspective of quality, it is high. The authors proposed a causally debiased time-aware recommender framework to learn user preference and considered both item-level and the time-level biases. And the authors theoretically prove that the method is unbiases to the ideal objective. Additionally, the data analysis is thorough, well-executed, and adequately supports the conclusions drawn.
Clarity
In this paper, the introduction provides a clear overview of the research topic and objectives, and the body sections are logically organized. And the language used in this paper is clear and easy to understand. Besides, some key concepts are well explained.
Significance
The work in this paper is of great significance. Firstly, CDTR outperforms other forecasting models. And then, the unmeasured confounders can be captured by a sensitivity analysis method.
And then, WINNET harvests the high computational efficiency for other forecasting models and make full use of the correlation between period trend and oscillation.

Cons
The framework of CDTR is needed.
Although the article provides detailed theoretical analysis, there are still certain parts that are difficult to understand and require multiple readings to grasp the main points.
In the experiment, I noticed that in some cases, the performance of DR-UM is not the best. More explanation is needed.
Please pay attention to the layout of the paper, for example, avoid having only one word per line.

**Questions:**

Using different strategies, S1 or S2, the metrics is quite different. The metrics using the strategy of S1 is lower than that using the strategy of S2. So, why still using S1 and S2?

There are differences between experimental results and theoretical demonstrations. Have you considered the reason for the difference?

**Reviewer Confidence:**

3: The reviewer is confident but not certain that the evaluation is correct

**Scope:**

4: The work is relevant to the Web and to the track, and is of broad interest to the community

---

### Official Review · Reviewer_rdLC · 2023-11-24

**Novelty:** 4
**Technical Quality:** 5

**Review:**

The paper addresses the issue of bias in time-aware recommender systems. The proposed solution uses a causal graph to identify biases at both item and time levels and employs the inverse propensity score (IPS) method, extended to a doubly robust method, to optimize for an unbiased learning objective. Additionally, the framework includes a sensitivity analysis method to better handle unmeasured confounders.

Strength:
- The authors provide a robust theoretical analysis connecting the proposed method with the ideal learning objective.
- The paper introduces a framework that addresses both item and time-level biases.
- Open sources implementation code to support reproducibility.

Weakness:
- The concept of the methods introduced appears to be a reiteration of ideas previously explored in other recommendation scenarios, lacking significant novelty.
- While the authors suggest that their framework is applicable to both explicit and implicit feedback, the evaluation is limited to explicit feedback. Notably, the CoNCARS base model, originally designed for implicit feedback, is not tested in this context. The authors should encompass implicit feedback metrics, such as NDCG or Hit Rate, to substantiate these claims.
- It would be great if the authors could include more recent models for debiasing or unobserved confounders.

**Questions:**

- The authors mention that the symbols are presented in Table 1 (in section 3), but the Table 1 is the performance table.
- Is there any particular reason not to include Amazon in ablation studies? It is also great to include some ablation studies with S2 time processing to show the consistency.

**Reviewer Confidence:**

2: The reviewer is willing to defend the evaluation, but it is likely that the reviewer did not understand parts of the paper

**Scope:**

3: The work is somewhat relevant to the Web and to the track, and is of narrow interest to a sub-community

---

### Decision · Program_Chairs · 2024-01-22

**Decision:**

Accept

**Comment:**

The paper proposed a debiased time-aware recommender framework by incorperating a crafted causal graph. While the studied problem is interesting, the paper indeed needs improvment on evaluation and paper presentation.